# A Novel Catheter Distal Contact Force Sensing for Cardiac Ablation Based on Fiber Bragg Grating with Temperature Compensation

**DOI:** 10.3390/s23052866

**Published:** 2023-03-06

**Authors:** Yuyang Lou, Tianyu Yang, Dong Luo, Jianwei Wu, Yuming Dong

**Affiliations:** 1School of Physics and Electronic Engineering, Chongqing Normal University, Chongqing 401331, China; 2Opto-Electronic Engineering and Technology, Shenzhen Institute of Advanced Technology, Chinese Academy of Sciences, Shenzhen 518055, China

**Keywords:** contact force sensing, cardiac ablation, fiber bragg grating, temperature compensation

## Abstract

Objective: To accurately achieve distal contact force, a novel temperature-compensated sensor is developed and integrated into an atrial fibrillation (AF) ablation catheter. Methods: A dual elastomer-based dual FBGs structure is used to differentiate the strain on the two FBGs to achieve temperature compensation, and the design is optimized and validated by finite element simulation. Results: The designed sensor has a sensitivity of 90.5 pm/N, resolution of 0.01 N, and root–mean–square error (RMSE) of 0.02 N and 0.04 N for dynamic force loading and temperature compensation, respectively, and can stably measure distal contact forces with temperature disturbances. Conclusion: Due to the advantages, i.e., simple structure, easy assembly, low cost, and good robustness, the proposed sensor is suitable for industrial mass production.

## 1. Introduction

Compared with traditional open surgery, minimally invasive surgery has the advantages of small incisions, less pain, and faster recovery. Atrial fibrillation ablation is a common minimally invasive medical procedure, in which an extremely common surgical catheter must be used. This catheter has a strong need for integrated contact force (CF) sensing capabilities, as it is closely related to the success of the procedure [1,2,3]. The poor success rate of radio frequency (RF) catheter ablation is mainly due to the inability to create transmural lesions. Recently, CF between the catheter tip and the myocardium has been identified as one of the important determinants influencing RF lesion size [1,2,3]. The continuous catheter CF monitoring is an effective means of balancing ablation effectiveness and safety [4]. CF-sensing catheters have improved recurrence rates [5], and significantly shorten the overall procedure including ablation time and fluoroscopy time compared to conventional catheterization [6]. CF-sensing catheters have also increased the safety of using RF-based ablation techniques [7], higher acceptance rates of complications, and lower rates of cardiac perforation [8,9,10,11]. Moreover, multicenter clinical studies have validated the effectiveness of CF catheters [12,13,14,15,16]. Therefore, catheter technology integrated with force-sensing ability has become the direction of the next generation of ablation catheters [4].

Fiber optic sensors have many unique advantages such as a strong resistance to electromagnetic interference, good compatibility with RF ablation devices, high measurement accuracy, great robustness, small size, good flexibility, high sensitivity and good physiological compatibility, which make them one of the important research directions for CF sensors [17,18,19]. The mechanical information is obtained by a microstructure design at the end of the optical fiber. Depending on the medical intervention needs, the commonly used optical fiber sensors are broadly classified into three categories: light intensity modulated, wavelength modulated, and phase modulated. For example, the force is detected by calibrating the relation between the applied force and the force-induced intensity change of the reflected light [20,21,22,23]. Fabry–Perot interferometry (FPI) is used to measure the force at the needle tip [24] and the cardiac catheter [13]. Fiber Bragg grating (FBG) is used for CF sensing at the distal end of the instrument [25,26,27,28,29,30]. Compared to light intensity modulation (susceptible to light source fluctuations) and phase modulation (limited in phase discontinuity), wavelength modulation has more consistency and higher sensitivity [31].

The accuracy of the sensor measurement, however, can be significantly impacted by the temperature coupling effect because FBG sensors are likewise sensitive to temperature. Thus, temperature compensation is necessary during force measurement [32]. The reference fiber grating approach and the symmetric temperature compensation strategy are now the main FBG temperature compensation methods used in catheter applications. For instance, Gan Lu et al. [25] developed a one-degree of freedom (DOF) extracorporeal tissue palpation force sensor with temperature compensation. They positioned the reference fiber grating next to the strain measurement fiber grating and sensed only the ambient temperature next to the strain measurement fiber grating without sensing the strain. The strain measurement fiber grating is subjected to both strain and temperature. The center wavelength shifts of both fiber gratings are subtracted from each other to obtain the wavelength shift of the fiber grating when it is subjected to strain alone, thus, eliminating or compensating the effect of temperature coupling. However, the shortcoming of this setup is that the reference fiber grating is loosely placed in the head and supported by only one end, which is not stable enough. Furthermore, its flexure design is a helical spring structure, which leads to its insufficient lateral stiffness and susceptibility to lateral forces. Another safety concern is the single helical connection between the distal and proximal portions, where structural failure at any location in this flexure could cause serious consequences [33]. Overall, the robustness of this configuration is not sufficient. Chaoyang Shi et al. [26] employed four fibers arranged equally spaced around the elastomer. The two opposing optical fibers along the direction of the flexure undergo tensional strain and compressive strain, respectively, both of which are of equal magnitude. By listing the corresponding equations, the strain term can be eliminated and the term affected by temperature only can be calculated, which is then used to obtain the compensated strain. However, this required one to use more fiber arrangements, and exposed the fibers to environmental interference. Moreover, it needs a more intricate structural design when applied to axial forces, which is difficult to be miniaturized.

To overcome the above problems, a novel temperature compensation design is proposed in this paper for the measurement of the distal contact force of AF ablation catheters. It consists of two elastomers and associated connectors. The two elastomers are designed with different flexural structures to separate the properties of their internal FBGs to compensate the temperature effect. For tight suspension, the fiber’s two ends are attached to the sides of the elastomer along its centerline, as shown in Figure 1a. Therefore, the internal FBG element can be directly compressed or extended in the axial direction to detect the strain introduced by the deformation of the two elastomers. This configuration is effective in avoiding FBG chirping failure and obtaining higher sensitivity compared to the direct FBG pasting approach. The structure has advantages in industrial mass production since it is straightforward, simple to construct, inexpensive, and robust.

## 2. Sensor Design and Sensing Principle

### 2.1. Design Requirements

The size of the sensor and the technical specifications are determined according to the specific requirements in the catheter ablation scenario. The design specifications are summarized in Table 1. The outer diameter of the sensor should be less than the outer diameter of the catheter [22] in order to achieve a minimized sensor structure for better integration in catheter ablation treatments; hence, the outer diameter of the sensor is designed to be 3 mm. The optimal CF during ablation procedures is 20–30 g [22]. A linear working range of up to 50 g and a force resolution of at least 1 g is therefore required [22].

### 2.2. Force-Sensitive Flexural Structure Design

The sensor prototype presented in this study is based on a two-segment multi-layer continuous beam-slots elastomer flexure structure designed to differentiate the strain sensitivity of the fiber inside the sensor, and thus achieve temperature compensation. The detailed configuration of the sensor of this design is shown in Figure 1a, which is mainly composed of a force-sensitive flexure and a double-FBG fiber. The flexure structure consists of two elastomers and blockers (sensor cap, connector, and sensor base). The fiber with two FBG segments is attached in the small hole of the blocker and is designed to be suspended tightly along with the desired flexure centerline. During the gluing of the fiber, some pre-tension is applied to the fiber to ensure a good linear response when the fiber is compressed. The two FBGs are positioned within the two hollow elastomers, as shown in Figure 1d. To control the influence of different FBGs on the sensing characteristics, similar FBG wavelengths of 1545 nm and 1555 nm are used. Regarding the selection of the FBG grating length, the decrease in FBG grating length would lead to a lower reflectivity and larger bandwidth, which in turn would increase the challenge of the demodulation. Therefore, we adopt 5 mm for all the grating lengths to ensure a certain size and facilitate the demodulation. We choose a multi-layer continuous beam-slots as the prototype of the force transducer, as it has good manufacturing feasibility and design flexibility, which is shown in Figure 1c. The overall structure is composed of biocompatible 304 stainless steel, and the elastomer is laser cut from a stainless steel tube that has a 3 mm outer diameter and 2.7 mm inner diameter. To achieve the difference in strain sensitivity, the beam slots of the elastomers 1 and 2 are arranged in 13 and 4 layers, respectively. The prototype sensor as well as the detailed dimensions are shown in Figure 1b,d. In summary, this design supports the decoupling of the force and temperature, is easy to machine and assemble, and has a simple structure with good robustness. The detailed working principle is described specifically in the next section.

### 2.3. Working Principle of Decoupling Force and Temperature

According to the working principle of FBG, when the sensor is subjected to a force in the axial direction, the specific process is shown below.

When the temperature and strain are varied simultaneously, the shift of the Bragg center wavelength is as follows:(1)$$\begin{array}{c} \begin{array}{c} \frac{\Delta \lambda }{\lambda }=\left(1-\rho_{e}\right)\varepsilon +\left[\alpha_{f}+\xi_{f}\right]\Delta T, \end{array} \end{array}$$
where Δλ is the Bragg wavelength shift, λ is the Bragg center wavelength, $\rho_{e}$ is the effective photoelastic coefficient of the sensing fiber, ε is the strain of the FBG due to a axial force, $\alpha_{f}$ is the thermal expansion coefficient of the sensing fiber, $\xi_{f}$ is the thermo-optic coefficient of the sensing fiber, and ΔT is the amount of ambient temperature change of the object under measurement.

When the thermal expansion effect of the object used to hold the fiber is considered, the equation can be written as
(2)$$\begin{array}{c} \begin{array}{c} \frac{\Delta \lambda }{\lambda }=\left(1-\rho_{e}\right)\varepsilon +\left[\alpha_{f}+\xi_{f}+\left(1-\rho_{e}\right)\left(\alpha_{s}-\alpha_{f}\right)\right]\Delta T, \end{array} \end{array}$$
where $\alpha_{s}$ is the equivalent thermal expansion coefficients of the elastomer applied to the fiber.

As the strain generated by the first and second elastomer is different, the forces on the first and second grating are also different, hence, the central wavelength shift of the FBG1 and the FBG2 is related to the strain, as follows:(3)$$\begin{array}{c} \begin{array}{c} \frac{\Delta \lambda_{1}}{\lambda_{1}}=\left(1-\rho_{e}\right)\varepsilon_{1}+\left[\alpha_{f}+\xi_{f}+\left(1-\rho_{e}\right)\left(\alpha_{s_{1}}-\alpha_{f}\right)\right]\Delta T, \end{array} \end{array}$$
(4)$$\begin{array}{c} \begin{array}{c} \frac{\Delta \lambda_{2}}{\lambda_{2}}=\left(1-\rho_{e}\right)\varepsilon_{2}+\left[\alpha_{f}+\xi_{f}+\left(1-\rho_{e}\right)\left(\alpha_{s_{2}}-\alpha_{f}\right)\right]\Delta T, \end{array} \end{array}$$
where $\Delta \lambda_{1}$ and $\Delta \lambda_{2}$ are the Bragg wavelength shifts of the FBG1 and FBG2, respectively, $\lambda_{1}$ and $\lambda_{2}$ are the Bragg center wavelengths of the FBG1 and FBG2, respectively, $\rho_{e}$ is the effective photoelastic coefficient of the sensing fiber, $\varepsilon_{1}$ and $\varepsilon_{2}$ are the strains of the FBG1 and FBG2 due to axial forces, respectively, $\alpha_{f}$ is the thermal expansion coefficient of the sensing fiber, $\xi_{f}$ is the thermo-optic coefficient of the sensing fiber, $\alpha_{s_{1}}$ and $\alpha_{s_{2}}$ are the equivalent thermal expansion coefficients of the first and second elastomers applied to the fiber, respectively, and ΔT is the amount of ambient temperature change of the object under measurement.

The stress is given by
(5)$$\begin{array}{c} \sigma =\frac{F}{A}, \end{array}$$
and Hooke’s law is given by
(6)$$\begin{array}{c} \sigma =E\varepsilon , \end{array}$$
the relation between strain and force can be obtained by
(7)$$\begin{array}{c} \varepsilon =\frac{\sigma }{E}=\frac{F}{AE}, \end{array}$$
where σ is the stress, ε is the strain, *E* is the modulus of elasticity, *F* is the applied axial force, and *A* is the cross-sectional area of the fiber.

When the sensor is subjected to a force *F* in the axial direction, the internal forces on the FBG1 and FBG2 are $F_{1}$ and $F_{2}$, respectively, and the corresponding strain $\varepsilon_{1}$ and $\varepsilon_{2}$ are shown as follows:(8)$$\begin{array}{c} \varepsilon_{1}=\frac{F_{1}}{AE}=\frac{K_{F_{1}}F}{AE}, \end{array}$$
(9)$$\begin{array}{c} \varepsilon_{2}=\frac{F_{2}}{AE}=\frac{K_{F_{2}}F}{AE}, \end{array}$$
where $K_{F_{1}}$ and $K_{F_{2}}$ are the force transfer coefficients of the optical fibers within the first and second elastomer, respectively.

Substitute Equation (Equation 8) in Equation (Equation 3) and substitute Equation (Equation 9) in Equation (Equation 4); the relation between force and Bragg wavelength shift is given as follows:(10)$$\begin{array}{c} \begin{array}{c} \Delta \lambda_{1}=\frac{\lambda_{1}\left(1-\rho_{e}\right)K_{F_{1}}}{AE}F+\lambda_{1}\left[\alpha_{f}+\xi_{f}+\left(1-\rho_{e}\right)\left(\alpha_{s_{1}}-\alpha_{f}\right)\right]\Delta T, \end{array} \end{array}$$
(11)$$\begin{array}{c} \begin{array}{c} \Delta \lambda_{2}=\frac{\lambda_{2}\left(1-\rho_{e}\right)K_{F_{2}}}{AE}F+\lambda_{2}\left[\alpha_{f}+\xi_{f}+\left(1-\rho_{e}\right)\left(\alpha_{s_{2}}-\alpha_{f}\right)\right]\Delta T, \end{array} \end{array}$$
set
(12)$$\begin{array}{c} K_{1F}=\frac{\lambda_{1}\left(1-\rho_{e}\right)K_{F_{1}}}{AE}, \end{array}$$
(13)$$\begin{array}{c} K_{2F}=\frac{\lambda_{2}\left(1-\rho_{e}\right)K_{F_{2}}}{AE}, \end{array}$$
(14)$$\begin{array}{c} K_{1T}=\lambda_{1}\left[\alpha_{f}+\xi_{f}+\left(1-\rho_{e}\right)\left(\alpha_{s_{1}}-\alpha_{f}\right)\right], \end{array}$$
(15)$$\begin{array}{c} K_{2T}=\lambda_{2}\left[\alpha_{f}+\xi_{f}+\left(1-\rho_{e}\right)\left(\alpha_{s_{2}}-\alpha_{f}\right)\right], \end{array}$$

Equations (10) and (11) can be simplified as
(16)$$\begin{array}{c} \Delta \lambda_{1}=K_{1F}F+K_{1T}\Delta T, \end{array}$$
(17)$$\begin{array}{c} \Delta \lambda_{2}=K_{2F}F+K_{2T}\Delta T, \end{array}$$
translate Equations (16) and (17) into a matrix form as
(18)$$\begin{array}{c} \left[\begin{array}{c} \Delta \lambda_{1}\\ \Delta \lambda_{2} \end{array}\right]=\left[\begin{array}{cc} K_{1F} & K_{1T}\\ K_{2F} & K_{2T} \end{array}\right]\left[\begin{array}{c} F\\ \Delta T \end{array}\right], \end{array}$$
where $\left[\begin{array}{cc} K_{1F} & K_{1T}\\ K_{2F} & K_{2T} \end{array}\right]$ is the sensitivity matrix, $K_{1F}$ and $K_{2F}$ are the force sensitivity coefficients of the first and second gratings, respectively, $K_{1T}$ and and $K_{2T}$ are the temperature sensitivity coefficients of the FBG1 and FBG2, respectively.

Since the two FBGs are in the same environment, their perceived temperature variations are approximately equal, so Equations (16) and (17) can eliminate the temperature term and obtain the decoupled contact force, then the relation between the target contact force measured by the sensor and the wavelength shifts of the fiber grating can be established as
(19)$$\begin{array}{c} F=\frac{K_{2T}\Delta \lambda_{1}-K_{1T}\Delta \lambda_{2}}{K_{2T}K_{1F}-K_{2F}K_{1T}}. \end{array}$$

### 2.4. Numerical Simulation and Optimization

Numerical simulations are performed by using the ANSYS Workbench (ANSYS, Inc., Canonsburg, PA, USA) prior to machining the designed sensors. In order to validate the feasibility of the above design, a finite element analysis of the structure under the applied load is performed with the relevant parameters shown in Table 2. Since the catheter is subjected to greater forces, around 2 N during the actual operation, for example when the catheter passes through the hemostatic sheath that keeps the artery open, the simulation is performed under a 2 N force loading to ensure that the sensor is feasible during the procedure. The simulation results are shown in Figure 2. From Figure 2a, it can be observed that the external structure of the sensor produces an obvious compressive strain around its multi-layer continuous beam-slots. It can be observed from Figure 2b that a large strain difference is generated between the fiber region I where FBG1 is located and the fiber region II where FBG2 is located, verifying the effectiveness of the flexural structure on improving the strain difference between the two FBGs. The overall structure remained stable under a 2 N axial force, and no structural failure occurs.

The first stage is to guarantee that FBG2 can detect the wavelength change under an axial force of 0.01 N to fulfill the sensitivity, since its strain sensitivity inside the elastomer 2 is lower than that of FBG1. The resolution of the employed FBG interrogator is 1 pm, which translates to a fiber strain of around 1με. Therefore, the strain that FBG2 generates under an axial force of 0.01 N should be more than 1με to guarantee a response in the interrogator. To ensure that the difference in sensitivity between the two elastomers is large enough, region II should be made with as few sensitizing structures as possible. Therefore, a simulation was performed to sequentially increase the layer of the beam slot to obtain the strain of the fiber in region II. The results are shown in Figure 3.

The ultimate dimension of the flexure is established after multiple design iterations, and the flexure portion is made up of four layers of rectangular slots measuring 2.5 mm wide by 0.3 mm high with a 0.3 mm interlayer gap. Accordingly, the sensitivity can be improved by simply increasing the number of layers of beam slots on elastomer 1. Elastomer 1 increases the number of layers to 13 while ensuring a certain lateral stiffness.

A force within 0–2 N is applied to the flexure along the axial direction in the steps of 0.5 N. The strain distribution of the fiber along the axial direction under force stimulation and the relation between the applied force and the strain distribution along the fiber in the direction of the applied force are shown in Figure 4. It can be observed that there is a clear strain difference between the fiber in the I and II regions when subjected to the same force, and the strains in both suspended sections of the fiber are very stable due to the two-point adhesive method, which ensures the stable measurement of the FBG wavelength shift. The strain in the remaining part of the fiber is small and almost zero because the rest of the fiber is in the gluing region and is fixed in the hole slot. The feasibility is further verified.

In addition to studying the effect of force on the fiber strain, the thermal expansion effect from temperature changes may also affect the strain on the fiber differently. The strain on the fiber with different temperatures is then simulated. Figure 5 shows the strain distribution along the axial direction of the fiber under different temperatures (initial temperature of 25 °C) and the relation between the temperature and strain. It can be observed that the temperature change also has different degrees of effect on the strains in regions I and II. The strain value in region II is slightly higher than the strain value in the region I, which is the opposite of the force effect obtained previously. This is due to the fact that the stainless steel tube in region II has a higher stiffness, which is advantageous in transferring force to the fiber, i.e., a larger $K_{F_{2}}$ (>$K_{F_{1}}$).

As shown in Figure 6, the strains on FBG1 and FBG2 of the fiber with different forces and temperatures are applied simultaneously. It can be observed on both sets of dashed lines that both temperature and force effects on strain have a good linear relation and the change in force leads to a larger strain difference between FBG1 and FBG2. Moreover, the effects of the applied axial force and temperature on the fiber strain are the opposite. According to the simulation data, the force sensitivity coefficients $K_{1F}=297.6\;\mu \varepsilon /N$, $K_{2F}=99.0\;\mu \varepsilon /N$ for FBG1 and FBG2 are obtained. Moreover, the temperature sensitivity coefficients $K_{1T}=8.3\;\mu \varepsilon {/}^{\circ }C$, $K_{2T}=10.0\;\mu \varepsilon {/}^{\circ }C$ for FBG1 and FBG2 are obtained. The strain of 1με generated by the fiber causes a wavelength shift of about 1pm. The resolution of the sensor calculated by Equation (Equation 19) is 0.005 N.

Moreover, the dynamic performance of the sensor is evaluated using a modal analysis to determine its operating frequency range. Figure 7a–c shows the first-order mode to third-order mode, and their corresponding resonant frequencies are 1084.2 Hz, 1220 Hz, and 4791.4 Hz in order. Based on its modal analysis, a harmonic response analysis is subsequently performed with a constant 1 N force applied along the Z-direction at frequencies from 0 to 2000 Hz with an interval of 20 Hz. The response curves of the proposed sensor corresponding to the X, Y, and Z directions are shown in Figure 7d. When the excited frequency is in the range of 1000–1300 Hz, the sensor has a maximum vibration generating resonance. The amplitude of the response curve is close to a constant value in the range of 0–800 Hz, indicating that the sensor can effectively avoid resonance and achieve good dynamic response performance in this range. Since the human heart normally beats 60–100 times per minute (1–2 Hz) [22], this range can be used as a sensor operation frequency band to adequately meet the measurement requirements for cardiac catheterization.

## 3. Experiments and Results

In order to obtain the actual sensitivity matrix of the proposed sensor, the force and temperature effects need to be calibrated. As shown in Figure 8, the calibration system mainly consists of the designed sensor, an ATI 6-axis force-torque sensor (Nano17 SI-12-0.12, NC, USA), a NetBox (9105-NETBA, ATI, USA) used for data transfer, an FBG interrogator (SA-10002454; sampling rate: 100 Hz; resolution: 1 pm), a three-axis automatic displacement stage (KOHZU, Japan; resolution: 1 micron), a manual displacement stage, two sets of fixtures to fix the designed sensor and ATI sensor, and a computer.

The designed sensor is connected to the interrogator by an optical fiber to transmit the demodulated wavelength data to the computer, and the ATI sensor transmits the force value to the computer via Network Box. After a secondary development on the computer by using the LabView software platform, the wavelength data and force data are recorded at the same time to achieve the purpose of calibrating the sensor.

### 3.1. Static Calibration Experiments for Force

Static calibration experiments are carried out to obtain the force sensitivity coefficients $K_{1F}$ and $K_{2F}$ for FBG1 and FBG2. As shown in Figure 9, the force range rises from 0 to 2 N, and the displacement stage records data of every 1-micron movement to obtain the relation curve between applied force and wavelength shift. The fitted curve is obtained by using a linear fit, and the slopes of the two lines segments represent the force sensitivity coefficient of FBG1 and FBG2, respectively. The results show that the two FBGs have good linear responses to force, with R-squared values of 0.997 and 0.999 for FBG1 and FBG2, respectively. The force sensitivity coefficients $K_{1F}\;=\;217.8\;\mathrm{pm}/N$, $K_{2F}\;=\;178.4\;\mathrm{pm}/N$ for FBG1 and FBG2 are obtained.

### 3.2. Static Calibration Experiments on Temperature

Although the main purpose of this design is to differentiate the sensitivity response of the applied force on the FBG, the above simulation results demonstrate that there is also a non-negligible difference in its response to temperature, thus it is necessary to perform temperature calibration experiments on the designed sensor. An electrothermal blast oven (DHG-9070A) is adopted in the experimental platform to control the temperature around the sensor, and the wavelength value is transmitted to the computer by the interrogator. The corresponding temperature and center wavelength are recorded at the same time. As shown in Figure 10, the temperature increases from 26 °C to 55 °C, and the data are recorded at 1 °C intervals to obtain the relation curve of temperature and wavelength shift. The results demonstrate that the two FBGs also have good linear responses to temperature, with R-squared values of 0.997 and 0.995 for FBG1 and FBG2, respectively. The temperature sensitivity coefficients $K_{1T}\;=\;17.2\;\mathrm{pm}{/}^{\circ }C$, $K_{2T}=24.1\;\mathrm{pm}{/}^{\circ }C$ for FBG1 and FBG2 are obtained.

### 3.3. Simulation and Experiment Comparison

According to the sensitivity coefficient obtained from the above experiment, the resolution of the sensor can be calculated as 0.01 N, and the sensitivity is 90.5 pm/N. The comparison of the simulation and experiment results are shown in Table 3. It can be observed from the table that there are some deviations between the simulation and experiment results. This may be due to errors introduced by manual assembly, as well as stress relief generated at the fiber bonding, etc. There is also an error in the relation between the wavelength drift and strain in FBG.

### 3.4. Experiments for Investigation of Dynamic Performance

Dynamic experiments were carried out to further validate the effectiveness of the designed sensor and to investigate its dynamic performance. Since the contact force is generally controlled within 0.5 N during the actual atrial fibrillation ablation procedure [22], dynamic loads varying in the range of 0 to 0.5 N are loaded, preserving the wavelength data over time. The force applied to the sensor tip is calculated from the sensitivity matrix obtained from the above experiments and simultaneously compared with the data received by the ATI sensor. As shown in Figure 11a, the measurement results obtained by the two sensors are almost identical, demonstrating the excellent performance of the designed sensor with a stable and high-accuracy detection capacity. Using the ATI sensor measurements as a reference value, Figure 11b shows a scatter plot of the designed sensor and the ATI sensor, which directly illustrates the dynamic response errors with a correlation coefficient of 0.993 and root–mean–square error (RMSE) of 0.023 N, indicating a high degree of correlation between the designed sensor and ATI sensor.

### 3.5. Study of Temperature Compensation Properties

Experiments were designed to verify the temperature compensation performance of the sensor. The sensor is loosely placed (i.e., the applied load is 0 N as the reference) in a closed space made of acrylic plates. A hot air gun is inserted through a small hole reserved for continuously blowing hot air to apply a temperature disturbance to the environment around the sensor. An electronic thermometer inside the sealed box is used to detects temperature change near the sensor. After a period of heating, the thermometer showed a change from 25 °C to 37 °C, and we recorded the corresponding force from the ATI force sensor and the central wavelength shifts of the dual FBGs. The temperature-compensated force of the designed sensor is calculated from the data collected by both FBG1 and FBG2, as well as the uncompensated force calculated by FBG1 or FBG2 only. A comparison of the two results is shown in Figure 12, where it can be observed that the compensated results are closer to the reference value. Since this hot air gun’s heating is rough, the uneven temperature distribution at the start of the heating period causes a substantial inaccuracy, which decreases after the temperature become stable. The RMSE after temperature compensation is 0.04 N. The results reveal that the designed sensor is obviously temperature resistant.

## 4. Conclusions and Discussion

In this study, a one-dimensional miniaturized force sensor was designed for the distal force sensing of AF ablation catheters by integrating a dual elastic structure and dual FBGs. The two FBGs are inscribed in the same optical fiber and placed in two 304 stainless steel tubes. By using different layers of rectangular slots in the stainless steel tubes, the axial stiffness of the two elastomers can be differentiated. The overall structure of this sensor is simple and easy to assemble with good robustness. The force equation after temperature compensation is derived based on the sensing principle of FBG. Finite element simulation of the sensor was performed on ANSYS platform to analyze its static force, temperature response, and harmonic response characteristics, which is helpful to verify the feasibility and optimize the size of the designed sensor. The sensitivity matrix of the sensor was obtained by calibration experiments, as well as the related parameters. Finally, dynamic force loading experiments and temperature compensation experiments were conducted to further validate the feasibility and effectiveness of the designed sensor. The experimental results demonstrate that the proposed sensor has a good dynamic characteristic, an adequate temperature compensation effect, a resolution of 0.01 N, and a sensitivity of 90.5 pm/N, all of which match the technical requirements for AF ablation [22]. The design provides a new solution for distal axial force measurement of catheter-based instruments in minimally invasive surgextendery by eliminating the temperature interference. Future work will be focused on extending the force sensing in both X and Y directions to further extend the sensor DOF to three dimensions.

## Figures and Tables

**Figure 1 sensors-23-02866-f001:**
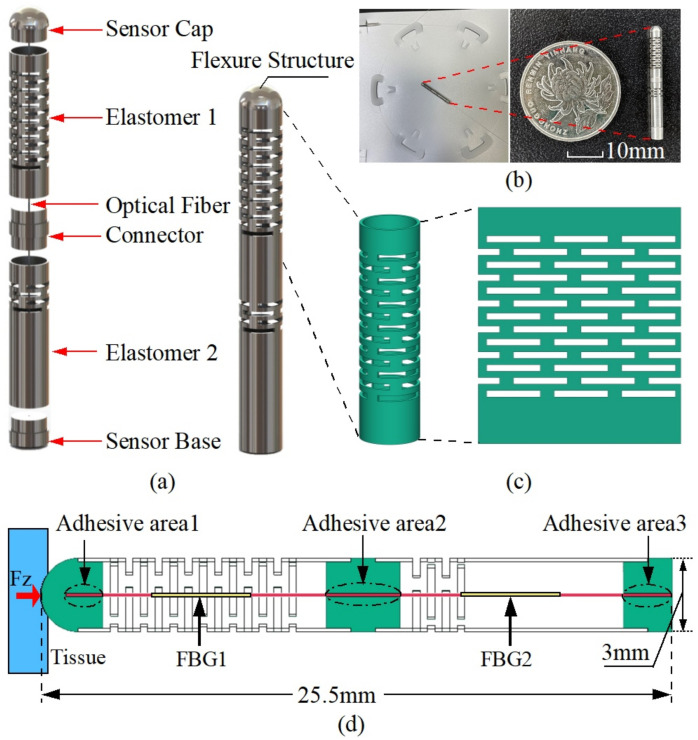
The schematic diagram of a dual elastomer fiber Bragg grating force sensor for cardiac catheterization. (**a**) Exploded view of the overall structure; (**b**) prototype; (**c**) multi-layer continuous beam-slots. (**d**) Detailed dimensions of the designed flexure and FBG arrangements.

**Figure 2 sensors-23-02866-f002:**
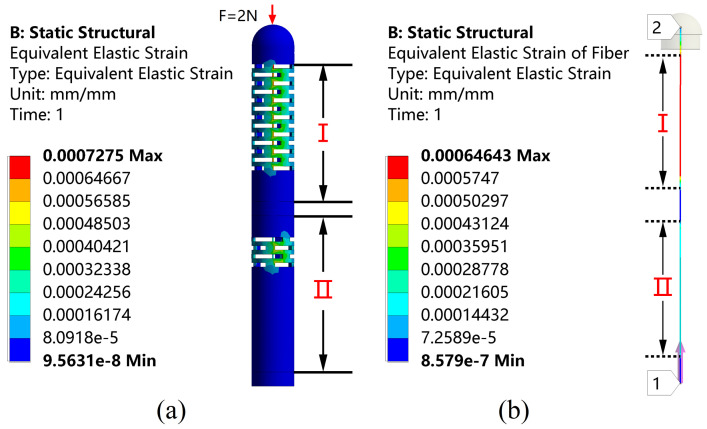
Strain distribution on (**a**) the overall structure and (**b**) the fiber under the axial force F stimulation.

**Figure 3 sensors-23-02866-f003:**
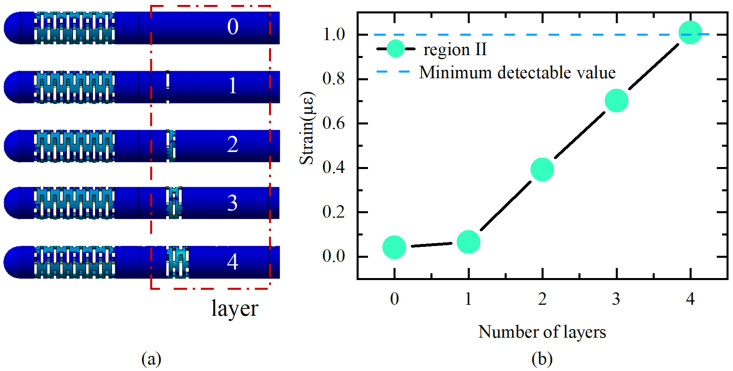
(**a**) Different structures; (**b**) strain of FBG2 under axial force of 0.01 N with different structures.

**Figure 4 sensors-23-02866-f004:**
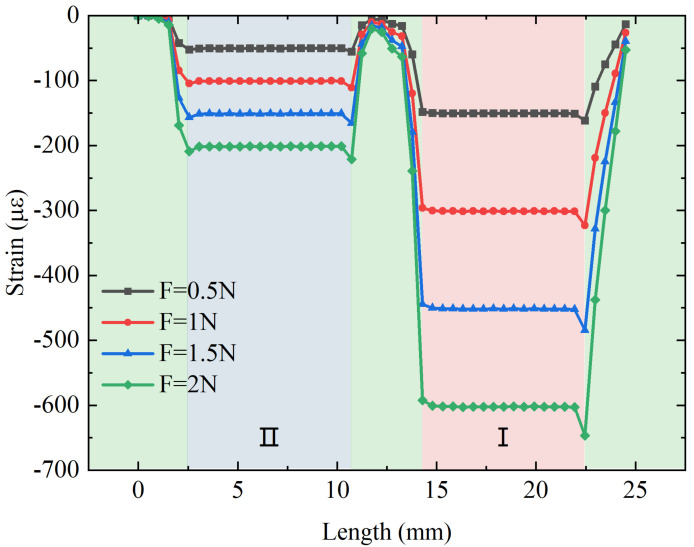
The relation between the applied axial force and the strain distribution along the suspended fiber.

**Figure 5 sensors-23-02866-f005:**
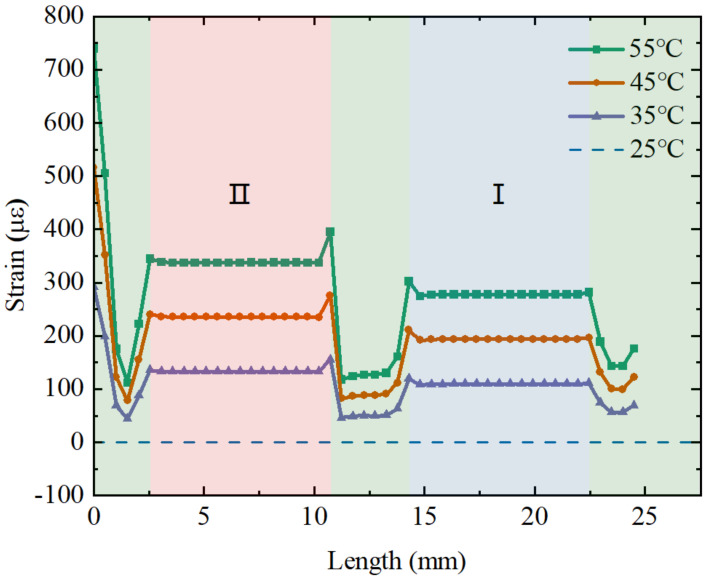
The relation between the applied temperature and the strain distribution along the suspended fiber.

**Figure 6 sensors-23-02866-f006:**
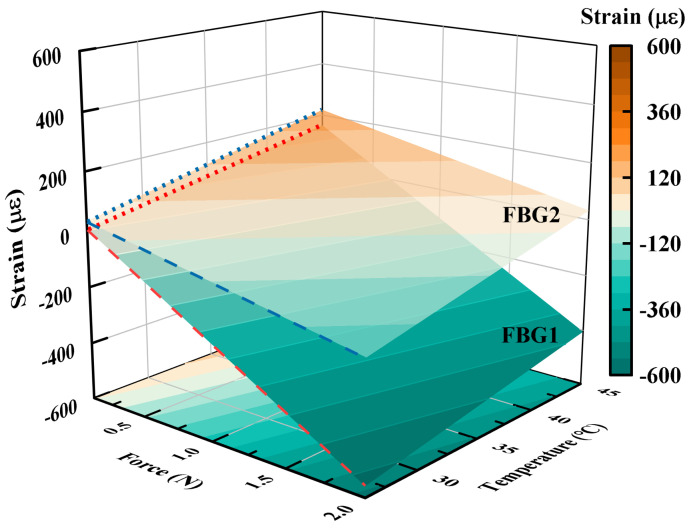
The relation between applied force and temperature and the strain of the two FBGs configured inside the sensor.

**Figure 7 sensors-23-02866-f007:**
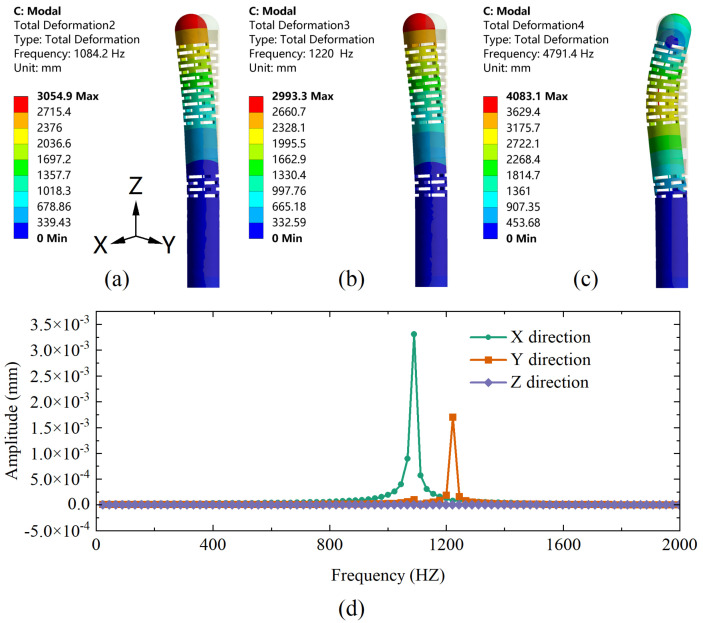
(**a**) First-order, (**b**) second-order, and (**c**) third-order modal analysis of the designed sensor. (**d**) The harmonic response curves of the designed sensor in X, Y, and Z directions under Fz conditions.

**Figure 8 sensors-23-02866-f008:**
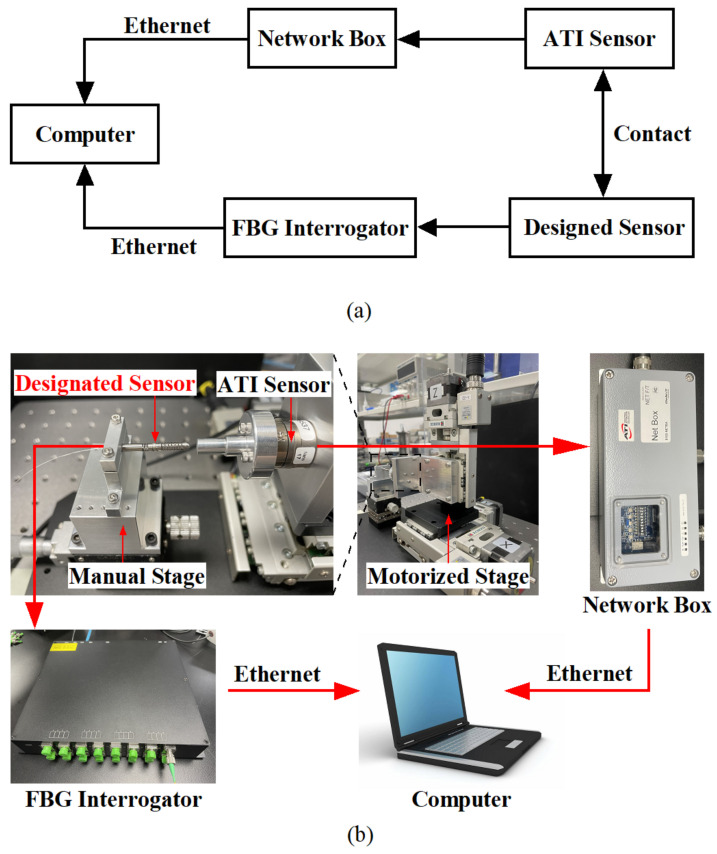
(**a**) Block diagram and usage flow of each equipment in the calibration experiment; (**b**) Experimental configuration for force calibration.

**Figure 9 sensors-23-02866-f009:**
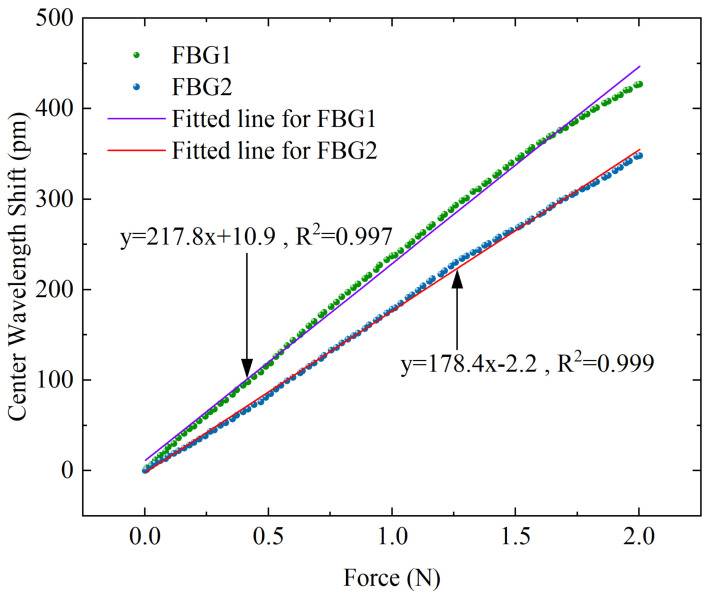
Corresponding response curves of center wavelength shift with different axial forces for the two FBGs.

**Figure 10 sensors-23-02866-f010:**
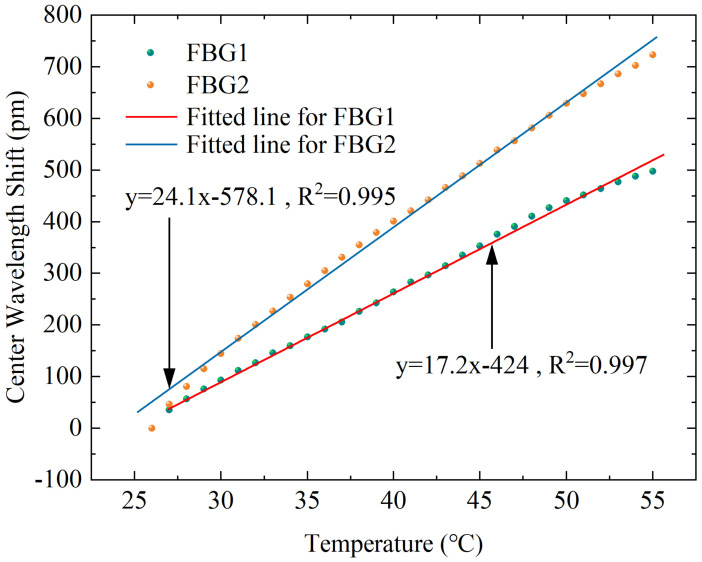
Corresponding response curves of the center wavelength shift with different temperatures for the two FBGs.

**Figure 11 sensors-23-02866-f011:**
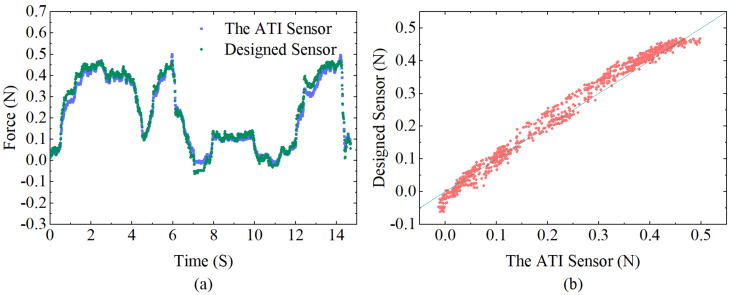
(**a**) Comparison between the calculated force of the designed sensor and the corresponding values detected from the ATI force sensor during dynamic loading. (**b**) Correlation diagram of the designed sensor and ATI force sensor data.

**Figure 12 sensors-23-02866-f012:**
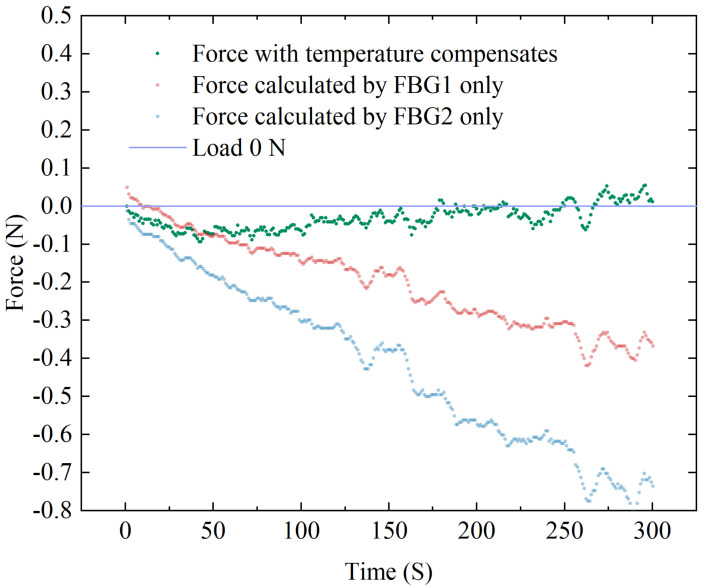
Temperature compensation experiment data of FBG sensor.

**Table 1 sensors-23-02866-t001:** Design specification of the proposed sensor.

Characteristics	Specifications
Sensor size	3 mm (9 Fr)
Working range	0–0.5 N (0–50 g of force)
Resolution	≤0.01 N

**Table 2 sensors-23-02866-t002:** Detailed parameters of the components of the sensor.

Specifications	Young’s Modulus (GPa)	Poisson Ratio	Density (kg/m^3^)
Flexure	195	0.3	7980
Optical fiber	72	0.17	2500

**Table 3 sensors-23-02866-t003:** Comparison of simulation and experiment performance value.

Performance	Simulation	Experiment
Force sensitivity coefficient (FBG1 and FBG2)	$K_{1F}=297.6\;\mu \varepsilon /N$ $K_{2F}=99.0\;\mu \varepsilon /N$	$K_{1F}=217.8\;\mathrm{pm}/N$ $K_{2F}=178.4\;\mathrm{pm}/N$
Temperature sensitivity coefficient (FBG1 and FBG2)	$K_{1T}=8.3\;\mu \varepsilon {/}^{\circ }C$ $K_{2T}=10.0\;\mu \varepsilon {/}^{\circ }C$	$K_{1T}=17.2\;\mathrm{pm}{/}^{\circ }C$ $K_{2T}=24.1\;\mathrm{pm}{/}^{\circ }C$
Resolution	0.005 N	0.01 N
Sensitivity	217.4 με/N	90.5 pm/N

## Data Availability

The experimental data are contained within the article.

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
