# Peer review of "A Novel Catheter Distal Contact Force Sensing for Cardiac Ablation Based on Fiber Bragg Grating with Temperature Compensation"

_sensors, 2023, doi:10.3390/s23052866_

Round 1

Reviewer 1 Report

This paper focuses on the problems of single variable, poor stability and high cost in the existing research, and proposes a temperature compensation design for measuring the distal contact force of AF ablation catheter, which has practical application value, but still has the following problems:

1. The title of section 2 is sensor design, but the content of the article contains relevant principle explanations. The content of the article should be consistent with the title. Please further refine the content structure and make a reasonable layout, which will make the article more convincing.

2. The proposed design should be compared and analyzed with the existing relevant research, which will make the proposed design more prominent and easy to judge. It is suggested to design tables for comparison, which will improve the preciseness and readability of the article

Reviewer 2 Report

The work presents a catheter distal contact force sensing device implemented in conjunction with two FBG sensors. The cascaded FBG sensors are differently packaged so that they have different responsivities to strain and temperature, which can be used to implement temperature-free force sensing in theory. FEM analysis has been provided to investigate the sensing structure as a whole in both static and dynamic conditions. The device may be a practically useful medical device, but the manuscript's current form is not publishable. I have the following specific points, addressing them might help improve the work.

1) In fig 1, it would be helpful if where and how the practical force(s) from the tissue may be applied to the sensing structure are denoted.  Both ends of the FBGs are glued to the housing, please make it clear in the schematic in Fig. 1d.

2) In practice, the FBGs should be pre-stretched so that they may respond to a compressing force as shown in fig 2, otherwise the fibers deflect easily when compressed.

3) Reference for eq 1 should be given. It seems more straightforward if KF2 and KF1 are called force transfer coefficients.

4) In the text above fig 5, it is stated that region 2 has more material thus a more pronounced thermal expansion effect. This statement might not be accurate, thermal expansion is not dependent on the volume of the material but the thermal property itself only. The more solid region 2 provides a higher stiffness which is advantageous in transferring force to the fiber, i.e., a larger KF2 (>KF1).

5) In practice, the dynamic force has a low frequency, see  fig 10a, thus the first order deflection frequency investigated in fig 6 is high enough and is more likely the practical deflection, thus an investigation on higher order modes might not be necessary.

6) A schematic in fig 7 would be clearer than the photos.

Reviewer 3 Report

Use of two FBGs for temperature-induced strain correction/compensation to accurately measure/isolate the strain due to the measurand of interest (axial force in this case) is a common method in FBG-based sensors interrogation. Here, the authors used the same scheme to measure the distal contact force of AF ablation catheters. The novelty here is the optimized design of the elastomers with different flexural structures so that the different strain coupling to two FBGs enables more accurate compensation for the temperature-induced strain. The Ansys simulation-based design optimization of the flexural structure and the experimental verification of enhanced sensitivity and resolution makes the work appealing to its targeted application.

Few questions that need to be addressed though:

1.     The elastomer structure was made of biocompatible 304 stainless steel and fabricated using laser-based process. One of the material requirements for the very force sensor used for targeted application is that it should not be magnetic/ferromagnetic for it to be also compatible with the MRI. Even though 304 SS is non-magnetic in normal operation, given that the flexural structure underwent through laser-cut process and potentially deformation, it could form some magnetic phases transforming it to a magnetic material. Could you comment on its magnetic properties, especially after the fabrication process whether there is any magnetic phase formation, whether the magnetic field-induced strain should also be considered given the high magnetic field strength used in MRI.

2.         The claimed resolution of 0.01 N is only in simulation. The experimentally achieved resolution is not 0.01. The RMSE in experimental data itself is 0.04 N. Please specify both resolutions, expected and measured, achieved in simulation and experiment respectively.

3.     In line 178 to 182, authors mention the optimized dimension of the flexural structure which was achieved after multiple iterations. Since this is the main design novelty of this work, I would suggest authors to include a plot (s) that shows strains values for the same 0.01 N force applied for various designs/dimensions considered for the optimization process and show that this structure is in fact optimized one.  

4.     In Fig.4, the legend for the corresponding colored lines in the plot shows that the strain is less at higher temperature ( 55 C) than at  lower temperature (40C, 35C). It should be other way round, higher strain at higher temperature. Please correct if it is a mistake or clarify otherwise.

Round 2

Reviewer 2 Report

No further comments.

Reviewer 3 Report

Authors have done excellent job in addressing reviewer's concerns and suggestion. As a result, the manuscript has been improved overall. I would recommend for its publication.